# Update on Current Imaging of Systemic Lupus Erythematous in Adults and Juveniles

**DOI:** 10.3390/jcm11175212

**Published:** 2022-09-03

**Authors:** Iwona Sudoł-Szopińska, Ewa Żelnio, Marzena Olesińska, Piotr Gietka, Sylwia Ornowska, Deborah Jane Power, Mihra S. Taljanovic

**Affiliations:** 1Department of Radiology, National Institute of Geriatrics, Rheumatology and Rehabilitation, 02-637 Warsaw, Poland; 2Connective Tissue Disease Department, Institute of Geriatrics, Rheumatology and Rehabilitation, 02-637 Warsaw, Poland; 3Clinic of Paediatric Rheumatology, National Institute of Geriatrics, Rheumatology and Rehabilitation, 02-637 Warsaw, Poland; 4Catalina Pointe Arthritis & Rheumatology Specialists, P.C. 7520 North Oracle Road, Suite 100, Tucson, AZ 85704, USA; 5Departments of Medical Imaging and Orthopaedic Surgery, University of Arizona, Tucson, AZ 85719, USA; 6Department of Radiology, University of New Mexico, Albuquerque, NM 87131, USA

**Keywords:** lupus arthritis, systemic lupus erythematosus, juvenile lupus erythematosus, lupus hand, rhupus, imaging, radiography, ultrasonography, magnetic resonance imaging

## Abstract

Systemic lupus erythematosus (SLE) is an autoimmune disease involving multiple organs and organ systems. Musculoskeletal (MSK) involvement is one of the most frequent and the earliest locations of disease. This disease affects joints and periarticular soft tissues, tendon sheaths and tendons, bones, and muscles. Multimodality imaging, including radiography, ultrasound (US), and magnetic resonance imaging (MRI), plays a significant role in the initial evaluation and treatment follow up of MSK manifestations of the SLE. In this paper, we illustrate MSK imaging features in three clinical forms of SLE, including nondeforming nonerosive arthritis, deforming nonerosive arthropathy, and erosive arthropathy, as well as the other complications and features of SLE within the MSK system in adults and juveniles. Advances in imaging are included. Conventional radiography primarily shows late skeletal lesions, whereas the US and MRI are valuable in the diagnosis of the early inflammatory changes of the soft tissues and bone marrow, as well as late skeletal manifestations. In nondeforming nonerosive arthritis, US and MRI show effusions, synovial and/or tenosynovial hypertrophy, and vascularity, whereas radiographs are normal. Deforming arthritis clinically resembles that observed in rheumatoid arthritis, but it is reversible, and US and MRI show features of inflammation of periarticular soft tissues (capsule, ligaments, and tendons) without the pannus and destruction classically observed in RA. Erosions are rarely seen, and this form of disease is called rhupus syndrome.

## 1. Introduction

Systemic lupus erythematosus (SLE) is a chronic, autoimmune, multisystem inflammatory disease which affects multiple organs and organ systems. The pathogenesis of this disease is only partially understood [1,2,3,4]. Antinuclear antibodies (ANA) are present in all patients [5], and almost one third of patients have antiphospholipid antibodies, which may cause thromboembolic complications [6]. SLE mainly affects women of reproductive age, and is characterized by heterogeneous degrees of severity as well as unpredictable flares and remissions [7].

Involvement of the skin and the MSK system are the most common manifestation of SLE, occurring in up to the 94% of patients [1,4,7]. Involvement of the MSK system is also often (in c.a. 50% of patients) the earliest manifestation of SLE [1,4,7,8]. Any joint may be affected, however, like in rheumatoid arthritis (RA), but most commonly, there is symmetric polyarthritis or oligoarthritis involving the hands, wrists, knees, and less frequently, the feet, ankle, shoulder, and elbow joints [2,6,9]. A hallmark of this disorder are articular deformities without fixed contractures [5].

The three main clinical manifestations of MSK SLE imaging features include: nondeforming and nonerosive arthritis (NDNE), deforming nonerosive arthropathy, called Jaccoud’s arthropathy (JA), and erosive arthropathy, called rhupus.

### 1.1. Nondeforming and Nonerosive Arthritis (NDNE)

The incidence of arthritis in patients with SLE of different races ranges from 69% to 95% [2]. About 78% of patients with lupus have arthritis at the initial diagnosis, and about 58% of patients with SLE relapses have active arthritis [2]. The typical manifestation of SLE arthritis is symmetrical, classically described as nonerosive, migratory, and reversible polyarticular inflammation. Most commonly, metacarpophalangeal (MCP), proximal interphalangeal (PIP), distal interphalangeal (DIP) joints, as well as knees and shoulders, are involved (2, 1). Only Iagnocco et al. [7] found that foot joints were the most frequently involved. Clinically, symptoms of arthritis can last from several hours to several months.

Ultrasound (US) and MRI show effusions and synovial and/or tenosynovial hypertrophy. Power and color doppler US, including newer microvascular techniques and contrast-enhanced MRI, show active synovial inflammation [2] (Figure 1). Compared with RA, the swelling caused by effusion and synovial hyperplasia and vascularity in SLE arthritis is relatively light [2]. Pathology reveals widespread vasculitis affecting capillaries, arterioles, and venules, and—unlike pannus in RA—villous hypertrophy of the synovium covered by fibrin and low-grade lymphoplasmacytic inflammatory cell infiltrates in the subintima [5].

### 1.2. Deforming Nonerosive Arthropathy/Jaccoud’s Arthropathy (JA)/Lupus Hand

This deforming arthritis without erosions, called Jaccoud’s arthropathy (JA), is pathognomonic for SLE [5]. As recently as the early 2000s, it was observed in up to 35% of patients [6], and now, as a result of more effective treatment, JA occurs far less frequently. This form of deforming arthritis resembles that observed in RA, but it is reversible [1,6]. This means that unlike in RA, where malalignments result mainly from intraarticular inflammation with destructive synovitis leading to articular bone damage, in SLE, the deformities occur secondary to a loss of support from the ligamentous and capsular structures around the joint, ligamentous laxity, and muscle contractures, and at least in the early stage of disease, they are completely reducible [5]. This reversible finding may be missed if only PA radiographs of hands are obtained (Figure 2). However, when hands are freed, different deviations become evident (thus the name reversible). They all result from inflammation followed by fibrosis of the periarticular soft tissues, including joint capsule, ligaments, and tendons without the pannus, which is classically observed in RA [1,10]. Deformities in the “lupus hand” include contractures in the MCP and interphalangeals joints, subluxation of MCP joints (aka ulnar deviation/ulnarization—the earliest pathology), swan-neck and boutonniere deformities, hitchhiker’s thumb, known as the “Z” deformity of the thumb, scapholunate dissociation, and ulnar translocation of the carpals [6,9]. Carpal instability as recently as the 1990s was found in 15% of SLE patients [11] and current data are not known. In feet, the deformities include contractures, the lateral deviation of metatarsophalangeal joints, hallux valgus and hammer toes [6,9]. Deformities may involve any other joint, such as the knees and shoulders [1].

Spronk et al. developed criteria to describe the severity of Jaccoud deformities [6], primarily based on the metacarpal axis deviation, and whether they are reversible or not based on the so-called Jaccoud’s index [12] (Table 1). Three different forms of deforming arthropathy in this disease were later proposed as follows: Jaccoud’s arthropathy (JA), mild deforming arthropathy (with less deformities), and the erosive form of SLE, called rhupus (Figure 3) [1].

### 1.3. Erosive Arthropathy, Called Rhupus

Rhupus syndrome (rhupus disease) is a rare form of SLE with malalignments and erosions, representing a type of SLE resembling RA [6] or an overlap syndrome, when RA coexists with SLE [1] (Figure 4). According to publications from the 1980s and 1990s, it accounts for c.a. 1–25% of SLE patients [6], and current data are lacking. Rhupus is indistinguishable from RA or RA patterns of psoriatic arthritis on imaging [10]. In all, articular erosions, tenosynovitis, tendon tears, and bone marrow edema (BME) are seen [9,10]. In SLE, Bywaters described hook deformities at the MCP joints [6], and Reilly et al. found ulnar styloid erosions [6]—in both locations, postulated due to adjacent tenosynovitis/tendinitis—so called compressive erosions [6].

The imaging features of three types of SLE described above are summarized in Table 2.

## 2. Update on Imaging of SLE on Radiography, Ultrasonography, and MRI

### 2.1. Arthritis

#### 2.1.1. Radiography

Arthritis in SLE is usually non-erosive in radiography, even in the 5–15% of patients with a long-standing disease who develop hand and foot deformities as hallmarks of Jaccoud’s arthropathy [14]. Radiographs are usually the first method in the imaging work-up. Bilateral radiographs are performed to evaluate the location and spectrum of imaging features and to differentiate between SLE and other rheumatic diseases [10]. SLE, like RA, involves the hand and wrist in a bilateral manner, whereas unilateral involvement is typical for Still’s disease [10].

In the case of hand and wrist involvement, the posterior–anterior (PA) view is supplemented with oblique radiographs in supination or the Nørgaard view (the ball-catcher view) [10]. This is especially appreciated in SLE, where, in addition to detecting more erosions in an additional view, reversible malalignment may not be apparent on a PA radiograph, corrected by the pressure of the hand against the radiographic cassette [5,9].

Radiographic features in SLE typically include [5,6,9]:Periarticular bone demineralizationPeriarticular soft tissue swellingSymmetrical polyarthritis, most commonly involving hands, wrists, knees, feet, and shouldersPreserved joint spacesDeforming, non-erosive arthropathy (Jaccoud’s arthropathy)Occasionally erosive arthropathy (rhupus)Acral sclerosis, acroosteolysis in some patients;Frequent osteonecrosis, most commonly of the femoral head, as the result of vasculitis and steroid therapy;Insufficiency fractures, possible due to disuse demineralization or osteopenia;Bacterial and fungal joint infections due to steroid administration and renal disease;Myositis, in a small number of patients, sometimes with the presence of calcifications;Uncommon spine manifestations, with atlanto-axial subluxation.

#### 2.1.2. Ultrasound and Magnetic Resonance Imaging

The imaging findings in US and MRI are nonspecific for many rheumatic diseases because synovitis, tenosynovitis, and BME can be seen in many of them, such as RA, juvenile idiopathic arthritis (JIA), PsA, and SLE [10]. In SLE, however, US and MRI improve understanding of the erosive disease and joint pathology [2] (Figure 2 and Figure 5). These advanced imaging methods identify articular and periarticular abnormalities in the early disease phase with high sensitivity, allowing for a more appropriate and accurate management and follow-up [3].

A literature search conducted by Ceccarelli et al. [3] in a number of databases on SLE imaging [3] found reported synovitis in almost 60% of patients with SLE, and the presence of erosions with a frequency ranging from 2% to 87% [3]. As expected, 87% of rhupus patients showed US-detected erosions. Patients with rhupus not only had a greater number of erosions, but also larger erosions than those with JA or NDNE arthritis, with prevalent involvement of the MCP joints [14]. US appeared more sensitive than conventional radiography in detecting bone erosions in SLE, although, comparing it with CT, the overall reliability of US in detecting bone erosions was moderate for rhupus syndrome (0.55) and JA (0.58) and poor for NDNE arthritis (0.10) [14].

Iagnocco et al., in a prospective study on 62 consecutive SLE patients, reported that the MTP joints were the most commonly affected site, with joint effusions, synovial hypertrophy, or synovitis (72.6 %) compared with the wrist, MCP, and PIP joints [7,15]. Synovitis was most commonly detected in MTP 2 and 4 [7,15]. The MTP synovial hypertrophy was present in 80% of the SLE cases, but with power Doppler signals seen in only 10% of cases, which was attributed to mechanical tissue irritation [7]. In the same study [7], at least a single US abnormality was detected in the majority of patients (87.1%), supporting the concept of a high prevalence of joint involvement in SLE [7]. However, only 40% of patients presented as having clinical features of joint involvement [7]. This dissociation between clinical and US imaging findings suggests a condition of subclinical inflammation.

The prevalence of subclinical synovitis in heterogenous studies, including—in most of them—consecutive, both symptomatic and asymptomatic, patients with lupus, ranges between 10% and 95% [8]. Honghu et al. [2], in a retrospective study, aimed to compare the role of hand and wrist US in diagnosing subclinical synovitis in SLE patients. Out of 41 included patients, 95.1% had joint abnormality. The most common US finding was synovitis (with a frequency of 19.7%), followed by tenosynovitis (7.1%), joint effusion (5.7%), and bone erosions (0.6%). Among the MCP joints, the most commonly affected joint was the MCP 3, followed by the MCP 4, MCP 2, MCP 1, and MCP 5 joints. Almost 32% of patients had wrist joint involvement, 4.8% had interphalangeal joint involvement of the thumb, 24.4% had PIP 2 involvement, 41.5% had PIP 3 involvement, 29.3% had PIP 4 involvement, and 7.3% had PIP 4 involvement. Twelve patients (29.3%) demonstrated knee joint involvement. Guillen et al. [8], in a multicenter cross-sectional study, found US features of arthritis in hands in about one-third of asymptomatic SLE patients with synovial thickening on gray scale images, while synovial hyperemia on power Doppler US was seen in one-fifth of the patients. The global prevalence of subclinical synovitis in the group with arthralgias was 38.2%. In the group without joint symptoms, that prevalence was 30%. No erosions or tendon ruptures were found in any patient [8]. Subclinical synovitis was reported in one study in 58.3% of SLE patients without joint involvement, through the US of the wrist and second and third MCP joints [8], and in 77% of SLE asymptomatic patients, with synovial hyperemia in 23% [8]. In another study identified in this cross-sectional study, 25% of patients with arthralgia of the hands had synovial effusion or hypertrophy. Iagnocco et al. [7] studied prospectively the foot and wrist of 62 consecutive SLE patients who were mostly asymptomatic during the US, demonstrating synovitis in about 20–32% of MCP and 8–11% of PIP joints.

The advantage of MRI is an ability to visualize BME and the additional location of inflammation, as well as the ability to detect more erosions with the benefit of multiplanar imaging and access to all bone surfaces. In SLE patients, BME at the wrist level, as found in Ceccarelli et al. from a literature search quoted earlier [3], ranges from 13% to 35% of the cases [3]. Ostendorf et al. [6] found that MRI of the hands of 14 SLE patients showed periarticular capsular swelling in all cases, joint effusion in 7, and mild synovial hypertrophy in 10 cases.

Recent application of US and MRI have probable influenced revision of the current Systemic Lupus International Collaborating Clinics (SLICC) criteria introduced in 2012 [3]. According to these criteria, patients with no swelling but positive US synovitis may benefit from escalating immunosuppressive therapy [16]. Conversely, a negative US may indicate that it is safe to taper corticosteroids, which is important given the toxicity of long-term corticosteroids [16]. Mahmoud et al. [16], in their longitudinal multicenter study involving 133 SLE patients, showed that 20% of patients with swollen joints upon clinical assessment do not have active synovitis.

Despite the clear advantages of US and MRI, SLE-specific quantification of inflammation is lacking. In 2003, early investigations with MRI noted the different features of SLE arthritis compared with RA, particularly the presence of edematous tenosynovitis and capsular swelling. Since that time, a few new evaluations of lupus arthritis using MRI have been reported [4]. One study [4] assessed the utility of Rheumatoid Arthritis Magnetic Resonance Imaging Score (RAMRIS) in the quantification of lupus arthritis scoring, as well as identified features of lupus arthritis that are incompletely captured by the RAMRIS. The authors compared patients who had objective findings of swelling upon clinical examination and low RAMRIS component scores, and found joint effusions that would not be scored as there was no enhancement with a gadolinium-based contrast and no synovial proliferation. They also noted tenosynovitis throughout the hand and fingers seen in these patients with SLE, even when tenosynovitis was minimal in the wrist. It is worth recalling that RAMRIS is the Rheumatoid Arthritis MRI Scoring system, and perhaps there is a need for an SLE MRI scoring system.

### 2.2. Tendons

Spontaneous tendon ruptures are a rare complication of SLE. US is the foremost imaging modality for tendon pathologies, since it is more sensitive than clinical examinations and MRI for detecting pathological structural changes within tendons [17]. The degree of tendon inflammation in rheumatologic diseases, namely tenosynovitis, paratenonitis, or tendinitis, as well as the extent of tendon damage, can be evaluated [17]. In such cases, US is able to depict tenosynovial effusion, tenosynovial thickening, and hyperemia, as well as tendon tears (partial, full-thickness, incomplete, or complete) [7,10]. In a literature search by Ceccarelli et al. [3], the prevalence of tenosynovitis ranged between 4% to 57%. Ostendorf et al. [6] in the MRI of the hands of 14 patients with SLE, found tendon sheath effusion (“edematous tenosynovitis”) in 42%.

Typical for SLE are spontaneous tendon ruptures (Figure 6, Figure 7 and Figure 8), most commonly of the patellar and Achilles tendons. In SLE, inflammatory changes as well as tears may be observed not only on the background of tenosynovitis, but also involving the tendon in the absence of tenosynovitis, even along the intramuscular segment [10] (Figure 8).

Contrary to RA, where tendon ruptures occur almost always in the hands as a result of tendinitis secondary to tenosynovitis and/or tendon tears against the eroded bone, in SLE, tears are most frequently seen in the lower limbs, affecting the patellar, Achilles, and quadriceps tendons, frequently associated with corticosteroid therapy with a superimposed mechanical component [1,10]. Tendon tears result mainly from corticosteroid’s antimycotic effect and fibroblasts’ inhibition of collagenase stimulus and consequential structural fiber disorganization [1]. Less frequently, tears are secondary to tenosynovitis, like in RA [1]. A systematic review using the MEDLINE, Scielo, and LILACS databases (1966 to 2009) demonstrated that almost one-third of the SLE patients with tendon ruptures also have JA; thus, this arthropathy may be recognized as a risk marker for tendon ruptures [18].

### 2.3. Myositis

In 5% to 10% of SLE patients, inflammatory myopathy is diagnosed based on laboratory findings, and muscular disease may be present in up to 50% of cases [1]. The main presenting symptoms are muscle weakness and myopathy detected by electromyography and muscle functional tests (i.e., MMT8—the Manual Muscle Test 8—diminished). Inflammatory myositis may be caused by the disease itself, but more commonly, SLE patients suffer from non-inflammatory myositis associated with the use of corticosteroids, chloroquine, or hydroxychloroquine [1]. The diagnosis is confirmed by muscular biopsy.

MRI is the most appropriate modality for the evaluation of muscle involvement, despite its low specificity, and US plays a complementary role [1,19] (Figure 9). MRI is used in the differential diagnosis and as a follow-up to a therapeutic response, and is useful to define the biopsy site [1,19]. Fat-saturated, fluid-sensitive MR sequences with long time until echo recovery are the most sensitive for identifying acute inflammation, manifesting as areas of high signals within the muscle [19]. Inflamed muscles also demonstrate contrast enhancement (Figure 9). In the chronic phase, involved muscles may undergo fatty infiltration with or without a loss of muscle bulk. These are seen on MRIs on T1-weighted sequences as areas of high signal intensity, replacing the normal intermediate signal of muscle fibers [19].

US may assist with the diagnosis and characterization of disease activity in inflammatory myopathies, with reported 82.9% sensitivity for detecting histologically proven myositis [17]. Inflammation and edema cause patchy or diffusely increased echogenicity of muscles, which may also appear swollen [17,19]. Increased vascularity on power Doppler may be seen [19]. In chronic diseases, the muscles appear atrophic with reduced volume and further increased echogenicity due to the progressive infiltration of fatty tissue [1,17,19]. In addition, by shear-wave elastography (SWE), US is able to evaluate muscle stiffness [20]. Di Matteo et al. [20] performed SWE on the quadriceps muscles in 30 SLE patients (without previous/current myositis or neuromuscular disorders) as well as 15 healthy subjects that matched the patients in age, sex, and BMI. SWE was significantly lower in SLE patients compared with the healthy subjects (1.5 m/s vs. 1.6 m/s respectively, *p* = 0.01).

### 2.4. Adipose Tissue and Lupus Panniculitis

Lupus panniculitis (LP), also referred to as lupus erythematosus profundus (LEP), is a chronic recurrent inflammation of the subcutaneous fat. It occurs in 1% to 3% of patients with SLE [21].

LP affects the deep dermis and subcutaneous adipose layer, and mainly involves the proximal extremities (lateral aspects of the arms and shoulders), thighs, buttocks, trunk, face, and scalp [21]. Patients usually present with persistent, often tender and painful skin lesions, or subcutaneous nodules, that range from 1 to 5 cm in diameter.

The imaging features of facial LP are extremely scarce in the literature (Figure 10).

US will likely show inflammatory changes and hyperemia along the involved and surrounding subcutaneous adipose layer. However, the main role of US is to exclude an underlying abscess, drainable fluid collection, or mass.

If US is unrevealing, MRI is the imaging modality of choice. The hallmark of LP on MRI is the loss of the normal T1 bright signal within the subcutaneous adipose layer. In active stages, a high signal intensity is seen on the fluid-sensitive sequences with bright, hazy enhancement on the post contrast T1-weighted images with fat saturation. It is important to distinguish LP from lymphoma, which unlike LP, does not track along the fatty tissue planes.

### 2.5. Bones

#### 2.5.1. Osteonecrosis

The prevalence of osteonecrosis (ON, avascular necrosis) in SLE according to the different authors ranges from 2% to 50% [1,11]. High-dose corticosteroid therapy (>20 mg/day) is undoubtedly the main determining factor, and AVN can develop within the first months of the initiation of treatment [1,6]. In SLE patients, ON occurs more frequently than in patients with other diseases that are treated alike [11]. Only 5–10% of patients are symptomatic [6]. The hips (femoral head accounts for >70%), knees, and shoulders are most commonly affected, but the ON of small bones can also occur [1,6].

Conventional radiography typically does not show any distinct pathology in the early phase of the disease, whereas the presence of subchondral sclerosis already infers irreversible osseous damage [1] (Figure 1). Abnormal findings in ON include a ‘crescent sign’, representing subchondral collapse; cyst-like or sclerotic changes; an abnormal contour; and the collapse of the femoral head with subsequent secondary degenerative changes [22]. MRI is the most sensitive modality in the diagnosis and quantification of the extent of ON [2,11,22]. Classic findings of medullary bone edema are observed on the fluid-sensitive sequences with fat saturation, with linear areas of low signal intensity inside representing the separation of normal and necrotic bones and the outer increased linear signal related to the vascularity of granulation tissue [11,22]. Whole-body MRIs may detect multifocal ON. Bone scans are less specific for the diagnosis of ON. Other limitations include the radiation dose, poor spatial resolution, and inability to quantify the lesion for prognostic purposes [22].

#### 2.5.2. Osteoporosis and Insufficiency Fractures

Many factors, such as renal failure, amenorrhea, early menopause, chronic inflammatory cytokines, and mainly, the chronic use of corticosteroids and anticoagulants, are involved in the genesis of osteoporosis in SLE [1]. Particularly in patients with SLE, the latter is a determining factor in the development of insufficiency fractures of the spine and other sites (particularly lower limbs). The prevalence of upper osteoporotic vertebral fractures in a study of Bultink et al. on 107 SLE patients was >20% [23].

Imaging modalities used in the assessment of osteoporosis include conventional radiography, conventional computer tomography (CT), dual-energy X-ray absorptiometry (DXA), quantitative CT, quantitative US, and MRI [24]. Today, radiography and DXA are the techniques of choice for vertebral fracture identification, whereas CT and MRI are used for characterization (dating and differential diagnosis) [25] (Figure 11).

#### 2.5.3. Calcifications and Acro-Osteolysis

Two types of periarticular calcifications occasionally occur in patients with SLE [6] (Figure 5). The first is dystrophic, like in dermatomyositis (DM), polymyositis (PM), systemic sclerosis, and mixed connective tissue disease (MCTD). The second is skin calcifications, like in calcinosis cutis [6].

Acro-osteolysis/acral resorption of distal phalanges and acral sclerosis may occur and are nonspecific for SLE. The acro-osteolysis is more frequently seen in scleroderma, hyperparathyroidism, and psoriatic arthritis (PsA) [6]. Acral sclerosis is seen in 10–12% of SLE patients, as found on radiographs by Braunstein et al. [26], but also may occur in RA, scleroderma, DM, sarcoidosis, and normal individuals [6].

## 3. Special Features of Juvenile Systemic Lupus Erythematosus

Childhood/juvenile systemic lupus erythematosus (jSLE, juvenile-onset SLE) has its onset before 18 years of age and accounts for up to 20% of SLE patients [11,27]. In contrast to the adult form, the adolescent onset of lupus is more aggressive and has worse outcomes, as found by a matched, multi-ethnic case–control study by Tucker et al. [28]. MSK symptoms were present in 60–90% of jSLE patients, depending on the study type, as analyzed by Levy et al. [24]. They include—like in adults—SLE-specific features (arthritis, tenosynovitis, myositis, osteitis, calcifications), disease-associated complications (osteoporosis and ON with insufficiency fractures), and infections [29].

### 3.1. Artricular and Periarticular Abnormalities

The most common, and usually the first, sign of arthritis is transient joint inflammation, most often involving the knees, ankles, hands, wrists, and less often, the elbow joints [11]. The arthritis is almost always nonerosive and nondeforming, migratory, and reversible [11,29].

Out of the three forms of deforming arthropathy concerning the hand, including mildly deforming arthropathy, Jaccoud’s arthropathy (JA), and rhupus hand, the last is not observed in children [11]. Moreover, JA is extremely rare, and epidemiology for the pediatric population is not available [11].

The imaging approach, like in adults, is based on radiography and US. Reversible deformities may be seen on PA and on the oblique views. US findings are similar to those seen in adults [11]. The prevalence of wrist synovitis found in a prospective study on 30 juveniles with SLE is 10.3%, as compared to 80% in adults [30].

Over time, ligamentous laxity and the instability of supporting structures may become fixed, resulting in contractures and muscle atrophy [11]. Dynamic US has the ability to differentiate contractures from tendons tears, although the latter is very rarely seen in children.

### 3.2. Myositis

Inflammatory myopathy is more common in children than adults. It is unclear if myositis is secondary to SLE, or is primary and only coexists with SLE [11]. MRI is the imaging method of choice with a supplementary application of US, including an assessment of vascularity and tissue elasticity with SWE.

### 3.3. Osteonecrosis and Insufficiency Fracture

In jSLE, ON is also observed, but less frequently than in adults [31]. Low bone mineral density related to corticosteroid use is frequent, and it is associated with an increased fracture risk, with a higher prevalence of upper spine vertebral fractures as well as at other sites, particularly the lower limbs [11]. However, risk factors for jSLE-related bone impairment are poorly understood [32]. Dual-energy X-ray absorptiometry (DXA) is the most widely used clinical tool for the assessment of bone density in children [33].

Diagnosis of ON and fractures is based on MRI in early stages (Figure 12) and radiography in advanced stages, like in adults.

## 4. Conclusions

Conventional radiography, US, and MRI have their specific applications in imaging of the MSK system’s involvement in SLE. Radiography is an important modality for differential diagnosis and treatment monitoring, including post-operative evaluation [3]. US is helpful in the case of an early disease or clinically evident synovitis. Both radiography and US are important tools in the differentiation between JA and RA, clinically presenting the same malalignments. MRI is useful in the assessment of soft tissue inflammation, BME, and erosive bone changes, and it is the imaging modality of choice in the early diagnosis of some complications such as ON, insufficiency fractures, or osteomyelitis.

Despite the clear advantages of US and MRI, an improved description and quantification of lupus arthritis is needed to move lupus treatment into an era of precision medicine [4]. In 2003, early investigations with MRI noted the different features of SLE arthritis compared with RA, particularly the presence of edematous tenosynovitis and capsular swelling. Since that time, a few new evaluations of lupus arthritis using MRI have been reported [4]. Still, contrary to RA, there is no scoring to measure the intensity of the inflammation of SLE on imaging. Further research is therefore needed to address specific imaging characteristics of SLE, in order both to increase the awareness of such findings among the radiological community and to be able to better serve clinicians in early diagnosis and treatment follow-ups.

## Figures and Tables

**Figure 1 jcm-11-05212-f001:**
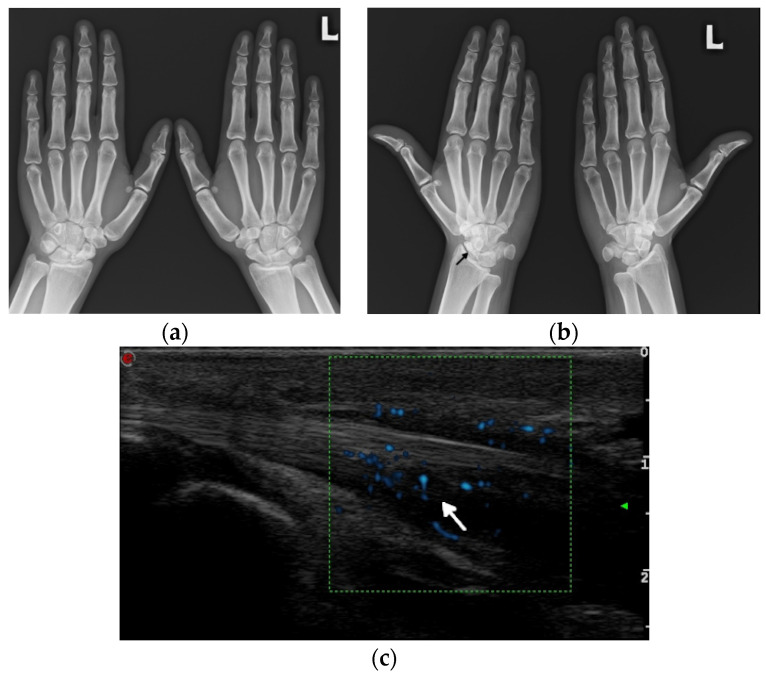
A 34-year-old female with systemic lupus erythematosus. (**a**) Posterior–anterior and oblique (**b**) radiographs of the bilateral hands show a non-united fracture of the right scaphoid waist, with increased sclerosis of the proximal pole and proximal waist consistent with osteonecrosis (arrow), possibly steroid-induced, and with no additional deformities. (**c**) Long-axis power Doppler ultrasound images at the volar aspect of the ring finger, (**d**) dorsal ulnar aspect of the wrist, and (**e**) volar aspect of the wrist show increased synovial vascularity involving the ring finger in (**c**) and the 6th extensor compartment tendon sheath in (**d**), consistent with tenosynovitis (arrows). Synovitis at the volar aspect of the radiocarpal and midcarpal joints without erosive bone changes is seen in (**e**) (arrowheads).

**Figure 2 jcm-11-05212-f002:**
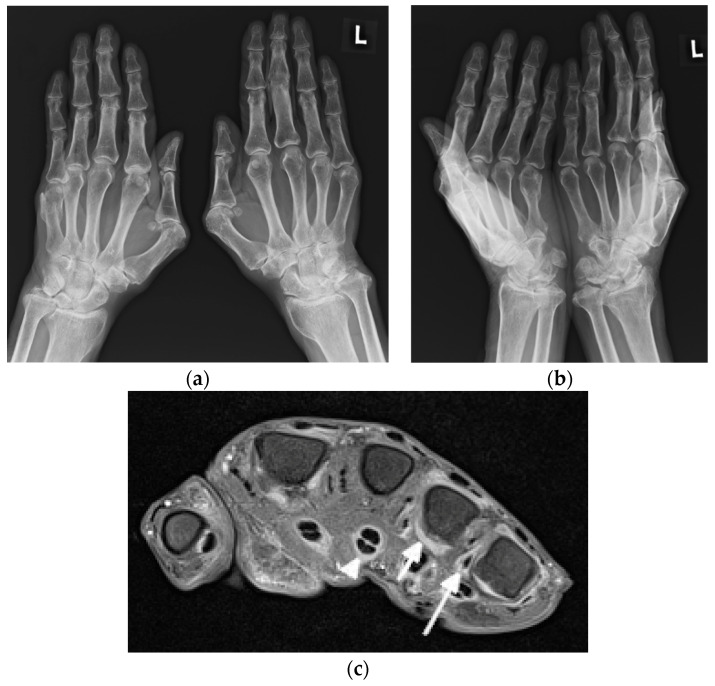
A 63-year-old male with systemic lupus erythematosus and Jaccoud’s deformities on clinical examination. (**a**) Posterior–anterior and (**b**) oblique radiographs of the bilateral hands show ulnar deviation of the lesser finger phalanges of the right hand, with malalignment of the 3rd–5th digits proximal interphalangeal (PIP) joints that are more apparent in (**b**). They also show contracture at the 3rd PIP joint of the left hand, osteoarthritis of the bilateral wrists and scattering of the metacarpophalangeal (MCP) joints, bilateral positive ulnar variance, and posttraumatic deformities of the bilateral distal radial metaphysis and of the distal right 5th metacarpal. (**c**) Axial postcontrast T1-weighted magnetic resonance image with fat saturation of the right hand shows MCP joints 2, 4, and 5 synovitis (short arrow pointing to MCP 4), MCP 2, 4, and 5 capsular enhancement (long arrow pointing to MCP 5), and 2–5 flexor tendons tenosynovitis (arrowhead pointing to middle finger flexor tendon sheath).

**Figure 3 jcm-11-05212-f003:**
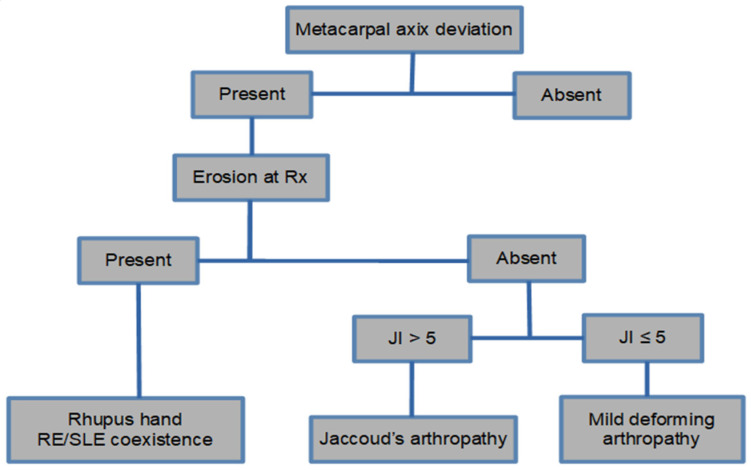
Algorithm with the forms of classification of joint involvement in SLE [13].

**Figure 4 jcm-11-05212-f004:**
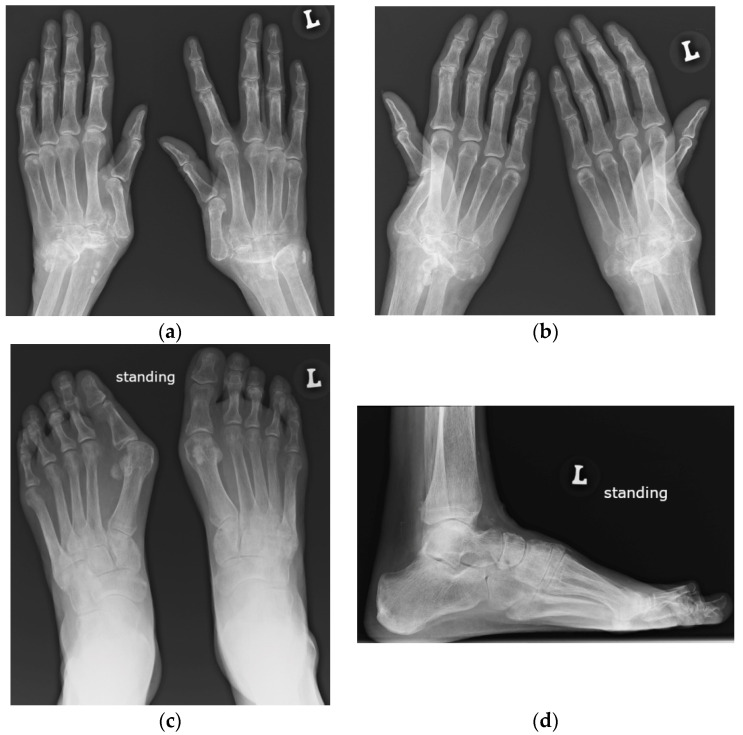
A 64-year-old female with SLE diagnosed in 1980 with Jaccoud’s arthropathy for hands and feet. (**a**) Posterior–anterior and (**b**) oblique radiographs of the hands show bone demineralization; soft tissue swelling at the wrists; periarticular calcifications; malalignment at the distal radioulnar, radiocarpal, and midcarpal joints; dislocation of the 1st carpometacarpal and subluxation of the 1st metacarpophalangeal joints; and joint space narrowing with cyst-like and erosive/destructive changes consistent with rhupus syndrome. Note the reversible contractures at the proximal interphalangeal joints, apparent in (**b**) and resolved or less apparent in (**a**). (**c**) Anterior–posterior standing radiograph of the bilateral feet and (**d**) lateral standing radiograph of the left foot show bone demineralization, bilateral hallux valgus deformities, and bilateral 2nd and 3rd hammer toes, as well as a right foot with moderate lateral subluxation.

**Figure 5 jcm-11-05212-f005:**
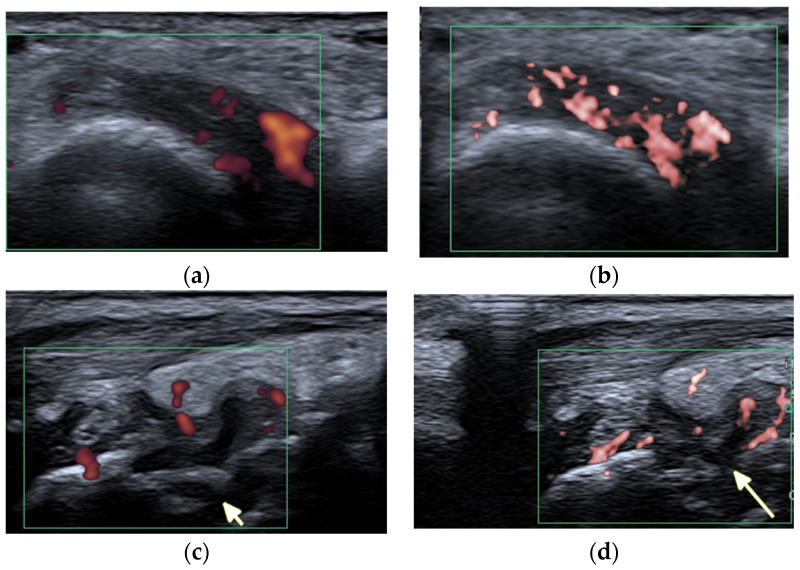
A 57-year-old patient with systemic lupus erythematosus. (**a**) Short-axis power Doppler and (**b**) superb micro-vascular imaging (SMI) ultrasound (US) images of the distal radioulnar joint. (**c**) Long-axis power Doppler and (**d**) SMI US images at the dorsal aspect of the radiocarpal and midcarpal joints show joint effusions and hyperemia consistent with synovitis with more vessels seen with SMI (**b**,**d**). Note an intra-osseous cyst-like change (short arrow) and cortical erosion (long arrow in (**d**)) in the carpal bone related to rhupus syndrome.

**Figure 6 jcm-11-05212-f006:**
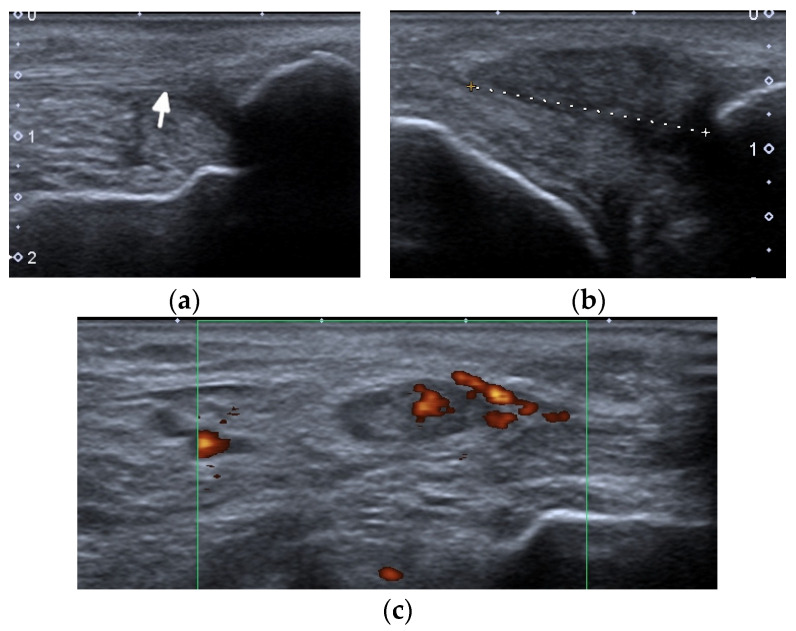
A complete spontaneous tear of the right posterior tibialis tendon (PTT) 2 cm above the medial malleolus in a 46-year-old female with systemic lupus erythematosus. (**a**,**b**) short-axis gray-scale ultrasound (US) images of the medial aspect of the bilateral ankles show the normal echogenic fibrillar appearance of the healthy left PTT (**a**, arrow) compared to an enlarged, torn right hypoechoic PTT in the same region between calipers in (**b**). (**c**) Short-axis power Doppler US image of the affected right side shows hyperemia in the PTT tendon stump, with additional hyperemia in the tendon sheath consistent with tendinopathy and tenosynovitis. Two tiny red dots at the periphery of green Doppler box represent normal vessels.

**Figure 7 jcm-11-05212-f007:**
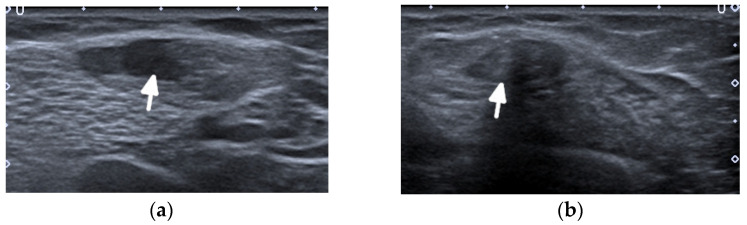
A complete tear of the proximal part of the bare tendon of the distal biceps brachii bilaterally in the same patient as in Figure 6. Short-axis (**a**,**b**) and long-axis (**c**,**d**) gray-scale ultrasound images of the bilateral elbow/distal arms show rupturing of the bilateral distal biceps tendons at the level of the myotendinous junction with hypoechoic proximal stums consistent with tendinopathy (arrows). In (**c**,**d**), note the retracted bilateral biceps muscles.

**Figure 8 jcm-11-05212-f008:**
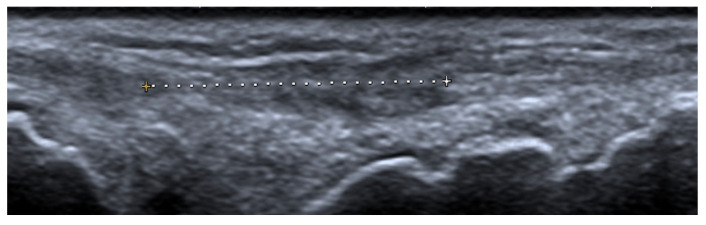
A long-axis gray-scale ultrasound image shows a complete tear of an extensor digitorum tendon of the middle finger at the level of the wrist (between calipers) in the same patient as in Figure 6 and Figure 7.

**Figure 9 jcm-11-05212-f009:**
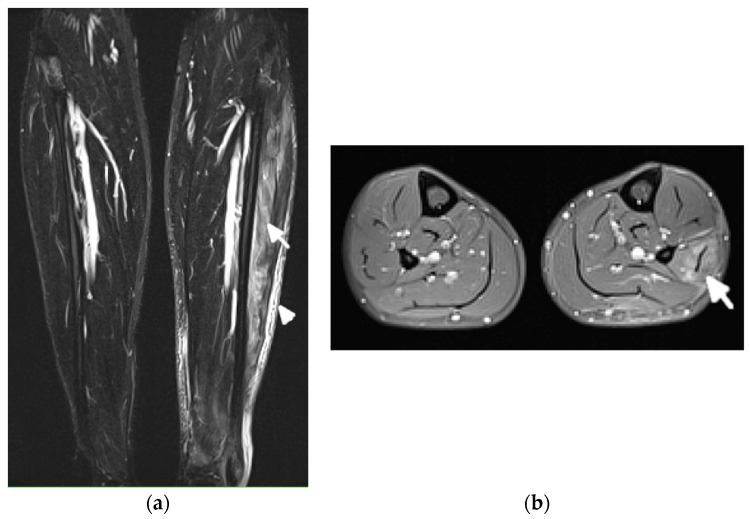
Magnetic resonance (MR) imaging of lower legs of a 34-year-old male with systemic lupus erythematosus with myositis. (**a**) Coronal T2 turbo inversion recovery magnitude and (**b**) axial postcontrast T1-weighted MR images with fat saturation show increased signals in the muscles of the lower left leg (arrows), especially peroneus brevis and the lateral head of gastrocnemius, and to a lesser extent, soleus musculature, with heterogeneous enhancement in (**b**) after the administration of a gadolinium base contrast agent. In (**a**), note the subcutaneous edema around the left lower leg (arrowhead).

**Figure 10 jcm-11-05212-f010:**
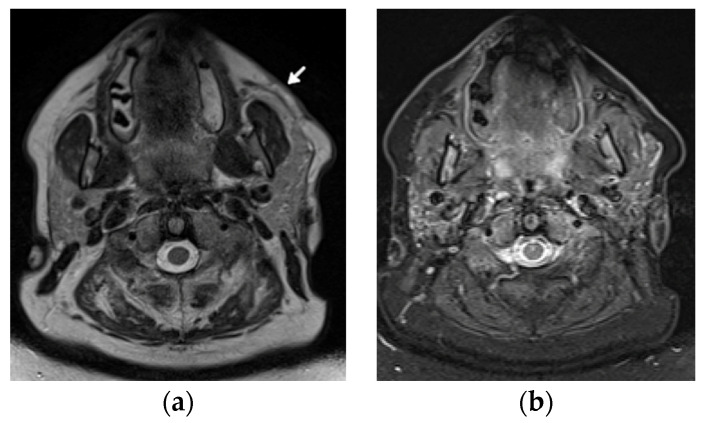
A 48-year-old female patient with systemic lupous erythematosus and a long history of lupus panniculitis and post-inflammatory atrophy of the subcutaneous tissues of the left cheek without active inflammation (arrow), seen in (**a**) the axial T2-weighted time spin echo and (**b**) T2 turbo inversion recovery magnitude MR images.

**Figure 11 jcm-11-05212-f011:**
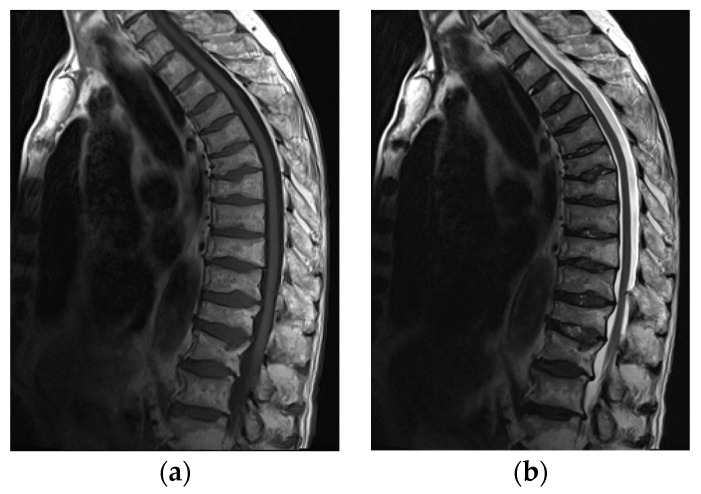
MRI of the spine in a 61-year-old patient with systemic lupus erythematosus with multiple compression fractures. (**a**) Sagittal T1-weighted and (**b**) T2-weighted Time Spin Echo MR images of the thoracic spine show a loss of height in nearly all thoracic and L1 and L2 vertebral bodies without significant bone marrow edema in (**b**). This is consistent with chronic compression fractures with associated disc dehydration and marginal endplate osteophytes, but without significant retropulsion.

**Figure 12 jcm-11-05212-f012:**
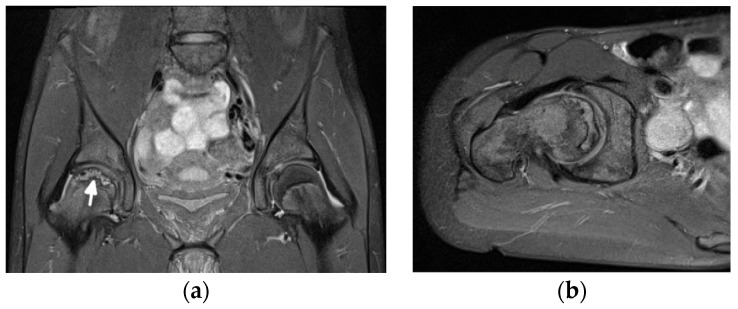
Magnetic resonance (MR) imaging of the femoral head osteonecrosis (ON) in a 14-year-old female with juvenile systemic lupus erythematosus. (**a**) Coronal and (**b**) axial proton density-weighted with fat saturation (right) MR images of the pelvis and right hip show a geographic serpiginous lesion in the right femoral head, with a bone marrow edema irregular contour and demarcation line consistent with ON (arrow).

**Table 1 jcm-11-05212-t001:** Diagnostic criteria according to Spronk. Jaccoud’s arthropaty is considered present if the scoring (Jaccoud’s index; JI) achieved is >5 [6].

Jaccoud’s Arthropaty Index (JI)	Number of Affected Fingers	Score
Ulnar deviation	1–4	2
	5–8	3
‘Swan-neck’ deformity	1–4	2
	5–8	3
Boutonniere deformity	1–4	2
	5–8	3
‘Z’ deformity of thumb	1	2
	2	3

**Table 2 jcm-11-05212-t002:** Imaging findings in three types of systemic lupus erythematosus on radiographs, ultrasound, and magnetic resonance imaging.

Magnetic Resonance	Ultrasonography	Radiography	
Imaging			
Effusions	Joint effusions	Periarticular bone demineralization	Nondeforming and nonerosive arthritis
Synovial thickening	Synovial thickening Hyperemia	Soft tissue swelling	
Postcontrast enhancement			
Periarticular BME			
Capsular inflammation (synovitis) Inflammation of the ligaments, tendon sheaths, tendons (tenosynovitis, tendinitis)	Capsular inflammation (synovitis) Inflammation of the ligaments, tendon sheaths and tendons (tenosynovitis, tendinitis)	Reversible malalignments	Deforming non erosive arthropathy/JA
Periarticular BME			
Erosions, synovitis and tenosynovitis, tendinitis, Periarticular BME	Erosions, synovitis, tenosynovitis, tendinitis	Erosions and malaligments	Erosive arthropathy/rhupus

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
