# Peer review of "Update on Current Imaging of Systemic Lupus Erythematous in Adults and Juveniles"

_jcm, 2022, doi:10.3390/jcm11175212_

Round 1
Reviewer 1 Report
Dear Authors,
You have written a review on imaging of musculoskeletal manifestations and complications of SLE based on 30 references in the literature. This summary will be of interest to both radiologists and clinicians who do not deal with this topic on a daily basis.
Unfortunately, there are numerous errors in craftsmanship, duplicate points in image descriptions, slipped image numbering, duplicate numbering in references, missing indentation in listings.
Abbreviations should be explained again in figures and tables.
Unfortunately, the work lacks structure.
Chapter 1 is Introduction and Arthritis of the Hand, Chapter 2 is Imaging, Chapter 3 is Complications, and Chapter 4 is Juvenile Lupus, 5 panniculitis, and 6 relevance.
It would be better to start in chapter 1 with a general introduction to SLE and the advantages of the 3 main types of imaging ultrasound, MRI and X-ray. Data and examples as well as relevance can then be provided to the various chapters, with arthritis in chapter 2, bone (osteoporosis, osteonecrosis) in chapter 3 tendons in chapter 4, muscle in chapter 5, adipose tissue in chapter 6, special features in jSLE in chapter 7, and an outlook in chapter 8.
For example, myositis is not a complication but a manifestation of SLE!
In the respective chapters, the DD of possible complications such as infection could be briefly mentioned, but the focus should be on the manifestations.
Please take into account the selection bias of the study when giving figures on the frequency. When retrospective radiographs are evaluated, the rate of deformity is of course higher than in cross-sectional studies.
We have a very large SLE cohort and I can assure you that the rate of Jaccaud arthropathy is certainly not 35%, as they write, probably more like 3.5%.
The frequencies should then also remain consistent, preferably with named method of estimation.
Experimental data like MRI imaging in fatigue do not belong in this clinical work, they can at best be named as an outlook.
The text should be shortened for the reader.
- Some anomalies are highlighted in the text.

Author Response
Answers to Reviewer 1
Thank you very much for your review.
- There are duplicate points in image descriptions, slipped image numbering, duplicate numbering in references, missing indentation in listings.
Ad.1. We have addressed all the above aspects.
- Abbreviations should be explained again in figures and tables.
Ad.2. This has been corrected.
- The work lacks structure. It would be better to start in chapter 1 with a general introduction to SLE and the advantages of the 3 main types of imaging ultrasound, MRI and X-ray. Data and examples as well as relevance can then be provided to the various chapters, with arthritis in chapter 2, bone (osteoporosis, osteonecrosis) in chapter 3 tendons in chapter 4, muscle in chapter 5, adipose tissue in chapter 6, special features in jSLE in chapter 7, and an outlook in chapter 8.
Ad.3. The structure of the paper has been corrected, many parts of the text were moved, some parts, one image were deleted, several new references were added.
- Experimental data like MRI imaging in fatigue do not belong in this clinical work, they can at best be named as an outlook.
Ad.4. This part has been deleted, as not strictly related to the paper aim, also in order to reduce the text.
5. Myositis is not a complication but a manifestation of SLE.
Ad.5. Thank you for this remark, this has been corrected.
- In the respective chapters, the DD of possible complications such as infection could be briefly mentioned, but the focus should be on the manifestations.
Ad.6. This aspect has been reduced, infections in adults and children were deleted.
6. Please take into account the selection bias of the study when giving figures on the frequency. When retrospective radiographs are evaluated, the rate of deformity is of course higher than in cross-sectional studies.
We fully agree with that suggestion. We went again through all quoted papers and added in particular sentences information on the study design, either retrospective, cross-sectional, review, if only such information was provided.
7. We have a very large SLE cohort and I can assure you that the rate of Jaccaud arthropathy is certainly not 35%, as they write, probably more like 3.5%.
Ad.7. Thank you for this comment which we agree since it reflects the current situation. Also in our practice, in our rheumatological center, JA, due to steroids and even 1 biologic available for SLE, is much less common. However the papers quoted in Weismann chapter (the latest from 2003) showed such high frequencies, and we can not find any original paper that shows reduced occurrence of JA. Nevertheless, we have modified that sentence.
- The frequencies should then also remain consistent, preferably with named method of estimation.
Thank you, as explained above- when such data were available, we have added an appropriate information.
- The text should be shortened for the reader.
Ad.9. The text has been reduced.
Thank you very much.
Reviewer 2 Report
The authors have made a good and comprehensive review of advances in imaging of systemic lupus erythematous. The authors have reviewed a large and up-to-date number of articles, most of them from the last 5 years. It is a subject in which there is still much to be done and that requires further studies. Despite the work done by the authors, I have doubts about the contribution to the scientific community of this review. The authors have extensively described each of the musculoskeletal affectations in SLE, but perhaps they should have focused more on the image itself.
I only have only two more comments:
1.- figure 1:
“and the 6th extensor compartment flexor tendon in (d) consistent with tenosynovitis (arrows).” I think it is a flexor tendon, not extensor compartment. The extensor compartment is on the dorsal aspect.“Synovitis at the volar aspect of the radiocarpal and midcarpal joints without erosive bone changes (arrowheads)”. ¿It is (e)?, please specify.
2.- “However, recent data indicated that, like in RA, this type of SLE complication may also be related to a prevalence of tenosynovitis with secondary tendinitis, focal tendon weakness, and later rupture [1].”
We can not speak about “recent data” when the study of Ribeiro [1] was published in 2011.
Author Response
Answers to Reviewer 2
Thank you very much for your comments. Below are our replies and explanations.
- What is the contribution of this paper to the scientific community of this review. The paper should be focused more on the image itself.
Ad.1. Systemic lupus erythematosus in a multiorgan disease, affecting numerous organs and systems, with a number of radiologic and clinical manifestations, and frequently misleading presentations, including on imaging. At the same time significant progress has been observed in the last years in the diagnosis and treatment of that disease. We think that this summary on imaging of musculoskeletal manifestations of SLE will be of interest to both radiologists and clinicians who do not deal with this topic on a daily basis. The disease might be misleading, whereas signficant progress has been observed in the last years in term of imaging, that can improve the diagnostic. We have reduced the text, and improved its structure (also addressing the other reviewer request), to make the paper more transparent and educational, and focused more on images.
- Figure 1: “and the 6th extensor compartment flexor tendon in (d) consistent with tenosynovitis (arrows).” I think it is a flexor tendon, not extensor compartment. The extensor compartment is on the dorsal aspect.“Synovitis at the volar aspect of the radiocarpal and midcarpal joints without erosive bone changes (arrowheads)”. ¿It is (e)?, please specify.
Ad.2. The images / location of lesions on the volar and dorsal sides are correct, there was a typo mistake in naming the (d) image, now it has been clarified; (e) was added.
- “However, recent data indicated that, like in RA, this type of SLE complication may also be related to a prevalence of tenosynovitis with secondary tendinitis, focal tendon weakness, and later rupture [1].” We can not speak about “recent data” when the study of Ribeiro [1] was published in 2011.
Ad.3. Thank you for this remark. This has been corrected.
Thank you.
Round 2
